# Neuronal reactivation during post-learning sleep consolidates long-term memory in *Drosophila*

**Ugur Dag[†], Zhengchang Lei[†], Jasmine Q Le[‡], Allan Wong, Daniel Bushey, Krystyna Keleman***

Janelia Research Campus, Howard Hughes Medical Institute, Ashburn, United States

**Abstract** Animals consolidate some, but not all, learning experiences into long-term memory. Across the animal kingdom, sleep has been found to have a beneficial effect on the consolidation of recently formed memories into long-term storage. However, the underlying mechanisms of sleep dependent memory consolidation are poorly understood. Here, we show that consolidation of courtship long-term memory in *Drosophila* is mediated by reactivation during sleep of dopaminergic neurons that were earlier involved in memory acquisition. We identify specific fan-shaped body neurons that induce sleep after the learning experience and activate dopaminergic neurons for memory consolidation. Thus, we provide a direct link between sleep, neuronal reactivation of dopaminergic neurons, and memory consolidation.

DOI: https://doi.org/10.7554/eLife.42786.001

**\*For correspondence:**
kelemank@janelia.hhmi.org

[†]These authors contributed equally to this work

**Present address:** [‡]Department of Biology, Brandeis University, Howard Hughes Medical Institute, Waltham, United States

**Competing interests:** The authors declare that no competing interests exist.

## Introduction

Animals guide their behavior in part based on the memories of past learning experiences. Temporally, these memories can be either short- or long-lasting; short-term memory (STM) shapes future animal behavior within seconds, minutes to hours after a learning experience, whereas long-term memory (LTM) holds the learned information for hours, days or even a lifetime. STM is thought to rely on protein synthesis-independent covalent modifications and changes in weight of existing synaptic connections, whereas LTM is believed to reflect protein synthesis-dependent structural changes in specific synapses (*Kandel, 2001*). Depending on the salience or duration of the learning experience, some STMs can be transformed into LTMs through the process of memory consolidation (*Dudai et al., 2015*).

Numerous studies in vertebrates suggest that sleep has a beneficial effect on memory in various learning tasks (*Marshall and Born, 2007*; *Smith, 2001*). Studies in humans and rodents have shown that an insufficient amount of sleep impairs cognitive functions such as memory formation and retention, while an ample amount of sleep after a learning experience supports memory storage (*Goel et al., 2009*; *Diekelmann and Born, 2010*). In addition, several studies have found that animals, including humans, exhibit an enhanced amount and quality of sleep after experiencing a novel or enriched environment (*Gais et al., 2002*; *Smith and Wong, 1991*; *Walker and Stickgold, 2006*). One hypothesis is that sleep plays an active role in consolidation of newly acquired memories into long-term storage, that are critical for the animal future actions. To date, however, a mechanistic understanding of sleep function in this selective memory consolidation remains elusive.

Over the last two decades, studies of sleep and LTM have expanded from the almost exclusive studies in mammals to other animals, including insects. Sleep is now well-documented in *Drosophila*. Sleep in *Drosophila* exhibits almost all the hallmarks of sleep in mammals, including homeostatic regulation, diminished behavioral responsiveness, and the existence of different sleep stages as

**eLife digest** Why do some memories fade after only a few seconds, whereas others last a lifetime? Studies suggest that part of the explanation has to do with sleep. Experiments in rodents show that neural circuits that are active during learning become active again when an animal sleeps. This process of reactivation, which may be akin to dreaming, helps strengthen specific memories and move them into long-term storage. But the complexity of the mammalian brain has made it difficult to pin down the underlying mechanisms.

One possible solution is to study the mechanisms in a simpler brain with fewer neurons, such as that of the fruit fly *Drosophila*. Dag, Lei et al. have now used molecular genetic tools to explore how sleep supports a specific type of learning in male fruit flies, called courtship learning. Female fruit flies that have recently mated will reject the courtship efforts of other males. A male fly that experiences repeated rejections therefore learns to avoid mated females in future. This type of memory can last for at least a day – a long time in the life of a fly.

Dag, Lei et al. show that males that experience repeated rejections subsequently spend more time asleep than control males. Preventing this sleep hinders the males from learning from their experience. But how does this process work? During sleep, specific dopamine neurons that were active during the learning episode become active once again. Blocking this reactivation prevents the flies from learning from their rejections. By contrast, artificially activating the dopamine neurons enables flies with only limited experience of rejection to learn to avoid mated females. Dag, Lei et al. show that neurons called vFB cells control this process. The vFB neurons both induce sleep and reactivate the memory-inducing dopamine neurons.

These findings in fruit flies thus reveal a direct causal link between sleep, reactivation of memory traces, and persistence of memories. They also show that fruit flies are a valid model for exploring the neural and molecular mechanisms connecting sleep and long-term memory.

DOI: https://doi.org/10.7554/eLife.42786.002

characterized by distinct electrophysiological signatures (*Dissel et al., 2015a*; *Yap et al., 2017*; *van Alphen et al., 2013*). Recently, a link between sleep and LTM has been established: sleep deprivation before and after learning impairs mnemonic performance (*Ganguly-Fitzgerald et al., 2006*; *Seugnet et al., 2008*; *Seugnet et al., 2009*), whereas sleep induction enhances memory formation and retention in wild-type (*Donlea et al., 2011*) and restores memory in memory mutant flies (*Dissel et al., 2015b*). Moreover, flies that were exposed to a socially enriched environment, including courtship learning, exhibit an increased amount of sleep (*Ganguly-Fitzgerald et al., 2006*).

Courtship conditioning has been extensively exploited to study both memory and sleep in *Drosophila*. Courtship conditioning is an ethologically relevant form of complex learning whereby male flies learn to associate the outcome of their own behavior with multisensory cues presented by females during courtship (*Ejima et al., 2005*). Naive *Drosophila* males eagerly court both virgin and mated females, which are generally receptive and unreceptive, respectively. However, after being repeatedly rejected by mated females, males become less likely to court other mated females (*Ejima et al., 2005*; *Keleman et al., 2012*). Like with other types of learning, the resulting memory can last from minutes to days depending on the training protocol.

We previously established that activity of the dopaminergic aSP13 neurons (DAN-aSP13s) is necessary and sufficient for STM acquisition via the dopamine receptor DopR1 in the γ neurons of the mushroom body known as Kenyon Cells (γKCs) (*Keleman et al., 2012*), a neuropil in the *Drosophila* central brain critical for memory formation (*Heisenberg et al., 1985*). Moreover, we recently demonstrated that the activity of the same DAN-aSP13s is also essential for the consolidation of courtship STM to LTM. This requirement is observed in a discrete time window after learning, and it is mediated via the dopamine receptor DopR1 in the γKCs (*Krüttner et al., 2015*).

In this study, we examine the mechanisms of DAN-aSP13 post-learning activation. We show here that DAN-aSP13s are activated during sleep after courtship experience that induces LTM. We present evidence that the specific class of sleep-promoting neurons in the fan-shaped body (FB), a neuropil previously implicated in sleep regulation (*Donlea et al., 2011*), activates DANs-aSP13 in the discrete time window after learning to consolidate courtship LTM.

## Results

### DAN-aSP13 neurons are activated during post-learning sleep

To investigate the role of sleep in post-learning activation of DAN-aSP13s, and hence LTM consolidation, we first asked whether DAN-aSP13s are active in freely behaving males after a prolonged experience with mated females, which induces LTM. To monitor activity of DAN-aSP13s for several hours in unrestrained males we employed a luminescence-based transcriptional reporter of neuronal activity (*Guo et al., 2017*). Using specific DAN-aSP13s GAL4 driver (*Aso et al., 2014a*) (*Figure 1—figure supplement 1A*), we expressed luciferase exclusively in DAN-aSP13s under the control of multimerized binding sites for the neuronal activity regulated gene Lola and FLP recombinase target sites (FRT) for its activity dependent cell specific expression (*MB315B-GAL4>UAS-FLP, Lola(FRT)stop (FRT)LUC*) (*Chen et al., 2016*). We measured luminescence as a proxy for neuronal activity in males that had undergone prolonged training with mated females and in naive males without prior courtship experience. As a control we used males after prolonged training with virgin females which does not induce memory when tested 24 hr later with mated females (our unpublished results). For training, single males were paired with a single recently-mated or virgin female for 6 hr. Afterwards, trained, naïve and control males were individually transferred to a 96-well luminescence reading plate. Luminescence measurements were taken every 15 min over 16 hr beginning at a common starting time for all males, at the hour seven from the onset of training. Notably, flies that had undergone training with mated females displayed a gradual increase of the luminescence signal, which was significantly different from naïve males between 7–10 hr from the onset of training (*Figure 1A*). In contrast, control males that had undergone training with virgin females also displayed an increase in luminescence however, it was identical to that in naïve males (*Figure 1—figure supplement 1B*). That specific time window of DAN-aSP13 activation after onset of training with mated females was preserved in males that were trained in a later circadian time during the day (*Figure 1—figure supplement 1C*). Together, these data show that DAN-aSP13s are activated in a specific time window after a learning experience that leads to LTM.

It was previously shown that flies that had been subjected to social enrichment, including courtship experience, display an increased amount of day time sleep (*Ganguly-Fitzgerald et al., 2006*). To determine whether *Drosophila* males sleep in the time period when DAN-aSP13s are activated, we measured the amount of sleep in males after prolonged courtship training and in naïve males. Sleep was analyzed with *Drosophila* Activity Monitors, and periods of inactivity lasting for at least 5 min were classified as sleep (*Shaw et al., 2000*; *Hendricks et al., 2000*). Both trained and naive wild-type males showed a significant amount of day time sleep, particularly during hours 7–10 when DAN-aSP13s are active. Importantly, males that had undergone training for LTM slept significantly more in this time window in comparison to naïve males (*Figure 1B*). Both groups displayed the same amount of sleep throughout the night.

Notably, enhancement of post-training sleep occurs between 7–10 hr from the start of training regardless of the circadian time of training. Males that were trained for LTM in the afternoon instead of the usual morning session, displayed an increased amount of sleep between 7–10 hr after onset of training (*Figure 1—figure supplement 1D*). They also had a normal LTM when tested 24 hr later (*Figure 1—figure supplement 1E*, *Figure 1—figure supplement 2A*). To evaluate their memory, we used automated video analysis to derive a courtship index (CI) for each male; CI is defined as the percentage of time over a 10 min test period during which the male courts the female. Memory is represented as a suppression index (SI), which is the relative reduction in the median courtship indices of trained versus naïve males: SI = 100*[1-CI$^{train+}$/CI$^{train-}$] (*Keleman et al., 2012*).

In contrast, males that were trained for 1 hr to induce STM did not exhibit an increased amount of sleep (*Figure 1—figure supplement 1F*). Together, these data suggest that only training that can induce LTM leads to sleep enhancement between 7–10 hr after the start of training.

To investigate whether the enhancement of post-training sleep is caused by learning or a prolonged intense activity such as courtship towards mated female, we monitored sleep pattern and courtship memory of mutants for the dopamine receptor DopR1 that do not form courtship memory due to an impairment in memory acquisition (*Keleman et al., 2012*). DopR1 mutant males, although underwent the same courtship experience and courted mated females as vigorously or more than the wild-type males during 6 hr training (*Figure 1—figure supplement 1G*), neither displayed an increased amount of post-training sleep (*Figure 1C*) nor formed LTM (*Figure 1—figure supplement*

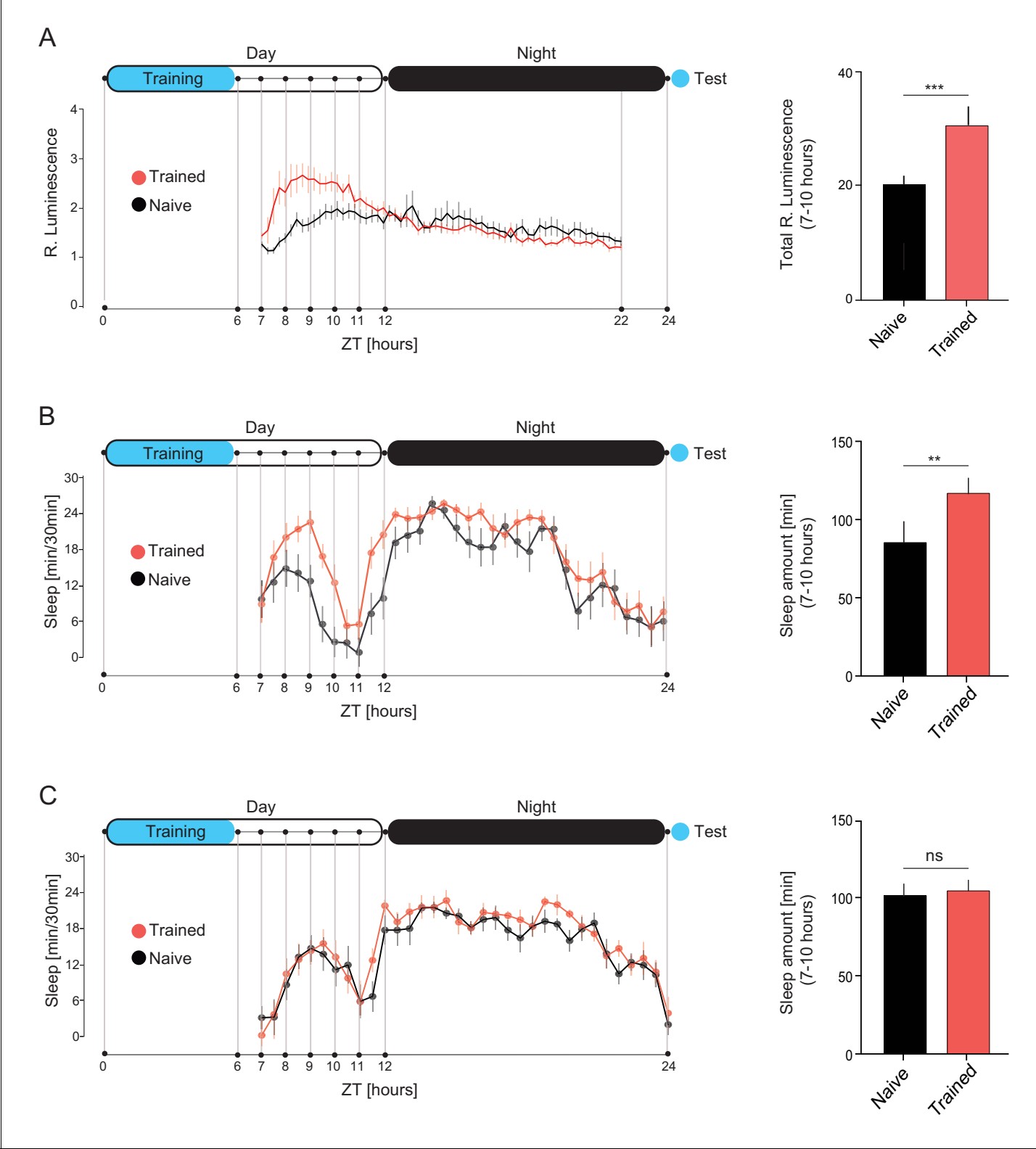

**Figure 1.** DAN-aSP13 neurons are activated during sleep. (**A**) (left) Luminescence of DAN-aSP13 neurons expressing Lola-LUC reporter (MB*315B-GAL4>UAS-FLP; Lola>stop>*LUC) normalized to luminescence of the genetic control (*UAS-FLP; Lola>stop> LUC*). Mean luminescence of the wild-type males trained with mated female in single pair assays as indicated (red, n = 42) and naïve males (black, n = 40) is shown as a solid line with SEM indicated as thin vertical lines. (right) Total luminescence in experienced and naïve males between 7–10 hr. P value is for Ho Luc$_{exp}$ = Luc$_{naive}$;

*Figure 1 continued on next page*

*Figure 1 continued*

***p<0.001. Student T-test. (**B**) (left) Sleep profile of the wild-type males that were trained for 6 hr with mated females (red, n = 16) and naïve males (black, n = 16). Sleep time was plotted in 30 min bins. (right) Total sleep of the experienced and naïve males between 7–10 hr. P value is for Ho Sleep$_{exp}$ = Sleep$_{naive}$; **p<0.01. Student T-test. (**C**)(left) Sleep profile of the *dopR1* mutants that were trained for 6 hr with mated females (red, n = 16) and naïve males (black, n = 16). Sleep time was plotted in 30 min bins. (right) Total sleep of the experienced and naïve males between 7–10 hr. P value is for Ho Sleep$_{exp}$ = Sleep$_{naive}$; ns p>0.05. Student T-test. Schematic of the experimental set-up in A, B and C indicates 12 hr light and dark periods (white and black areas) and time of training and test (blue shading).

DOI: https://doi.org/10.7554/eLife.42786.003

The following figure supplements are available for figure 1:

**Figure supplement 1.** Learning sufficient to induce LTM leads to the enhencement of the post-learning sleep and activation of DAN-aSP13 in the specific time window.

DOI: https://doi.org/10.7554/eLife.42786.004

**Figure supplement 2.** Courtship Indices (CIs) of males that had undrgone treatment according to *Figure 1—figure supplement 1E and H*.

DOI: https://doi.org/10.7554/eLife.42786.005

*1H*, *Figure 1—figure supplement 2B*). Together, these data show that learning is essential for the sleep enhancement in the specific time window after training.

## Sleep after learning is necessary for LTM consolidation

To determine whether post-learning sleep is necessary for LTM consolidation, we deprived flies of sleep during specific time intervals after training and tested their memory 24 hr later. Single wild-type males were trained with mated females for 6 hr and immediately after were exposed to intermittent gentle mechanical perturbation spanning 2 hr intervals. Naïve males were sleep deprived during the same time periods.

Control males that were trained with mated females and allowed to sleep, and males that were sleep deprived after training between 9–11 and 10–12 hr had normal SIs of approximately 30–40%. However, males that were deprived of sleep between 7–9 and 8–10 hr had SIs indistinguishable from 0 (*Figure 2A*, *Figure 2—figure supplement 2A*). Deprivation of sleep during the night did not impair LTM significantly, except for a small but statistically-significant effect between 14–16 hr (*Figure 2—figure supplement 1A*, *Figure 2—figure supplement 2E*). These results show that sleep deprivation between 7–9 hr after the start of training blocks LTM, and thus post-learning sleep is necessary for LTM consolidation.

Previously, we had shown that silencing of DAN-aSP13s in a discrete time window after training impairs LTM (*Krüttner et al., 2015*). To test whether the temporal effect of sleep deprivation mimics the effect of DAN-aSP13s silencing, we expressed specifically in DAN-aSP13s (*Figure 2—figure supplement 1B*) a temperature-sensitive inhibitory form of dynamin shibire (shi$^{ts}$) (*Kitamoto, 2002*) (*VT005526-LexA>LexAop-shi$^{ts}$*), which blocks synaptic transmission at 32°C but not at 22°C. We silenced DAN-aSP13s at 2 hr intervals after a 6 hr training session. Inhibition of DAN-aSP13s between 7–9 or 8–10 hr, but not between 9–12 hr, after the onset of training abolished LTM. Control males, in which DAN-aSP13s remained functional throughout the assay, had a normal SI between 30–40% (*Figure 2B*, *Figure 2—figure supplement 2B*). These results show that sleep and DAN-aSP13 activity are essential for memory consolidation in the same time window after training.

## Enhancement of sleep consolidates LTM

To investigate whether sleep can consolidate STM into LTM, we tested whether artificial enhancement of sleep would lead to memory consolidation. We trained males for 1 hr, which does not generate LTM (*Krüttner et al., 2015*), and then induced sleep at various time intervals afterwards. Since activation of the FB neurons induces sleep (*Donlea et al., 2011*), we expressed the thermosensitive cation channel TrpA1 (*Viswanath et al., 2003*; *Rosenzweig et al., 2005*) (open at 30°C and closed at 20°C) in FB neurons (*104y-GAL4 > UAS-TrpA1*) and monitored sleep for 24 hr. Consistent with previous reports, males in which FB neurons were activated slept significantly more than males that were kept at 20°C and the empty-GAL4 control males (*pBDP-GAL4 > UAS-TrpA1*) (*Figure 2—figure supplement 1C*).

To induce sleep precisely with 2 hr temporal resolution after a 1 hr training session, we used the optogenetic activator CsChrimson (*Klapoetke et al., 2014*). We found that activating FB neurons

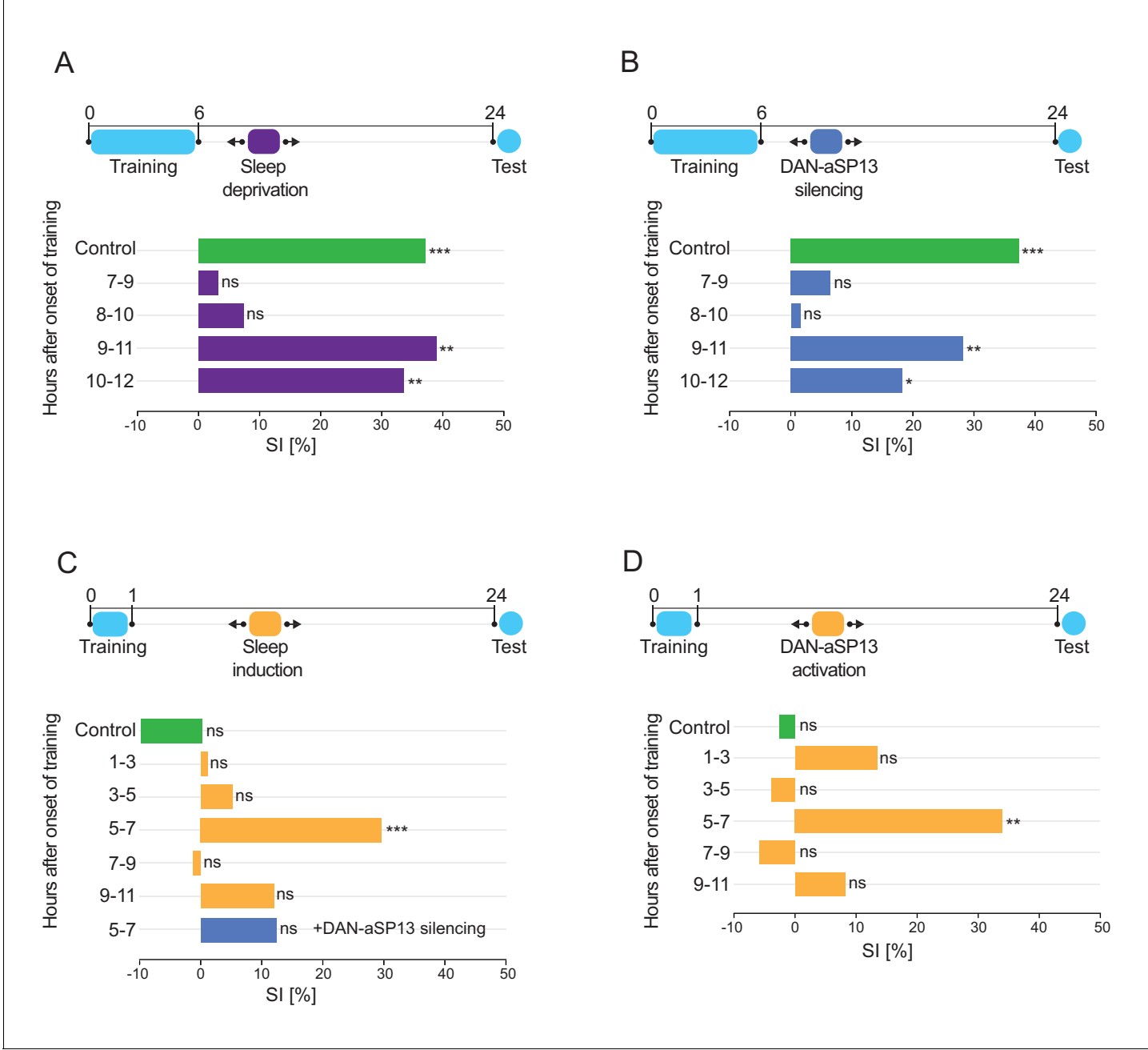

**Figure 2.** Sleep after learning is necessary and sufficient for LTM consolidation. (**A**) SIs of the wild-type males tested 24 hr after training for 6 hr with a mated female and sleep deprived at indicated time periods after training (dark purple bars). SI of the wild-type control males that were allowed to sleep (green bar). P value is for Ho SI = 0; ***p<0.001, **p<0.01, ns p>0.05. Permutation test. (**B**) SIs of males after training for 6 hr with a mated female and DAN-aSP13s silenced with shi^ts at indicated time periods (dark blue bars). SI of the control males with DAN-aSP13s active (green bar). P value is for Ho SI = 0; ***p<0.001, **p<0.01, *p<0.05, ns p>0.05. Permutation test. (**C**) SIs of males after training for 1 hr with a mated female and 104y neurons activated with csChrimson at indicated time periods (orange bars). SI of the control males with 104y neurons not activated (green bar). SI of the wild-type males with 104y neurons activated and DAN-aSP13s silenced between 5–7 hr after training (blue bar). P value is for Ho SI = 0; ***p<0.001, ns p>0.05. Permutation test. (**D**) SIs of males after training for 1 hr with a mated female and DAN-aSP13 neurons activated with csChrimson at indicated time periods (orange bars). SI of the control males with DAN-aSP13 not activated (green bar). P value is for Ho SI = 0; **p<0.01, ns p>0.05. Permutation test.

DOI: https://doi.org/10.7554/eLife.42786.006

The following figure supplements are available for figure 2:

**Figure supplement 1.** Activation of FB neurons induces sleep.

DOI: https://doi.org/10.7554/eLife.42786.007

*Figure 2 continued on next page*

*Figure 2 continued*

**Figure supplement 2.** Courtship Indices (CIs) of males that had undergone treatment according to *Figure 2* and *Figure 2—figure supplement 1A*.
DOI: https://doi.org/10.7554/eLife.42786.008

(*104y-GAL4 > UAS* CsChrimson) in the period between 5–7 hr after the onset of training led to a SI of ~ 30%, which is similar to that obtained after LTM training (*Figure 2C*, *Figure 2—figure supplement 2C*). In contrast, males activated at other time intervals or not activated at all failed to consolidate LTM and had SIs that were indistinguishable from 0. This temporal window was identical to that observed for DAN-aSP13s activation to consolidate LTM (*VT005526-LexA > LexAop*-CsChrimson) (*Figure 2D*, *Figure 2—figure supplement 2D*). When we activated FB neurons (*104y-GAL4 > UAS*-CsChrimson) while silencing DAN-aSP13s (*VT005526-LexA > LexAop-Shi^{ts}*) between 5–7 hr after the onset of training we did not observe LTM (*Figure 2C*, *Figure 2—figure supplement 2C*). Taken together, these data show that enhancement of sleep after a learning experience that induces STM can consolidate it to LTM, in a manner that requires activation of DAN-aSP13s. This LTM consolidation is a specific effect of DAN-aSP13s activation after training since activation of DAN-aSP13s (*VT005526-LexA > LexAop*-CsChrimson) in naïve males between 5–7 hr when they normally display a significant amount of sleep does not induce courtship suppression towards mated females and thus 'courtship LTM' (*Figure 2—figure supplement 1D*).

## FB neurons provide excitatory input to DAN-aSP13 neurons

To test whether FB activation excites DAN-aSP13s, we optogenetically activated FB neurons (*104y-GAL4 > UAS-Chrimson88*) and monitored calcium levels in DANs located in the protocerebral anterior medial (PAM) cluster in explant brains (*R58E02-LexA > LexAop-GCaMP6s*) (*Pfeiffer et al., 2008*; *Chen et al., 2013*). The PAM cluster of DANs consists of multiple neuronal classes, including a class of DAN-aSP13s, which is thought to respond to rewarding stimuli and promote their avoidance (*Aso et al., 2014b*) (*Figure 3—figure supplement 1A*). Since the expression pattern of 104y-GAL4 is not exclusive to the FB (*Donlea et al., 2011*), to activate FB neurons selectively, we used a digital mirror device (DMD) to target the illumination activating Chrimson to specific layers in the FB (*Strother et al., 2018*). 104y-FB neurons form three layers of projections: dorsal (dFB), medial (mFB) and ventral (vFB) (*Donlea et al., 2011*). To test the targeted illumination, we expressed in 104y neurons both Chrimson88 and GCaMP6s (*104y-GAL4 > UAS-Chrimson88 > UAS-GCaMP6s*) and selectively activated one of two layers of the FB (dFB or vFB) while monitoring calcium levels in the presence of tetradotoxin (TTX). TTX inhibits propagation of action potentials (*Boccaccio et al., 1999*), and thus calcium signals should only be observed in the projections located in the targeted layer. Upon activation of either dFB or vFB layers, a robust calcium response was observed only in the activated layer, implying that this local activation was highly restricted (*Figure 3A*).

We next activated all FB layers using targeted illumination and monitored calcium responses in DAN-aSP13s (*104y-GAL4 > UAS-Chrimson88, R58E02-LexA > LexAop-GCaMP6s*). Local activation of FB neurons elicited an excitatory response in DAN-aSP13s (*Figure 3B*). Given that the FB has a multilayered organization, we next aimed to identify the specific layer of the 104y-FB expression pattern that provides excitatory input to DAN-aSP13s. We individually activated all three 104y-FB layers and monitored calcium levels in DAN-aSP13s. Surprisingly, activation of the dFB layer, which has been recently implicated in homeostatic sleep regulation (*Donlea et al., 2011*), resulted in a mixture of small inhibitory and excitatory responses (*Figure 3C*), whereas activation of the vFB layer induced excitatory responses in DAN-aSP13s (*Figure 3D*). Activation of the mFB had no effect on DAN-aSP13 activity (*Figure 3—figure supplement 1B*). These data show that neurons that project to the vFB and potentially the dFB layers provide an excitatory input onto DAN-aSP13s and may thus have a role in LTM consolidation.

## vFB neurons promote sleep and activate DAN-aSP13s

To identify the specific neurons in the 104y-GAL4 line that project to the different FB layers, we performed multicolor flip-out experiments (*Nern et al., 2015*). We identified two distinct neuronal populations that project to either the dFB or vFB layers (*Figure 4—figure supplement 1A*). Next, we searched the Janelia (*Jenett et al., 2012*) and Vienna Tiles (VT) (*Tirian and Dickson, 2017*)

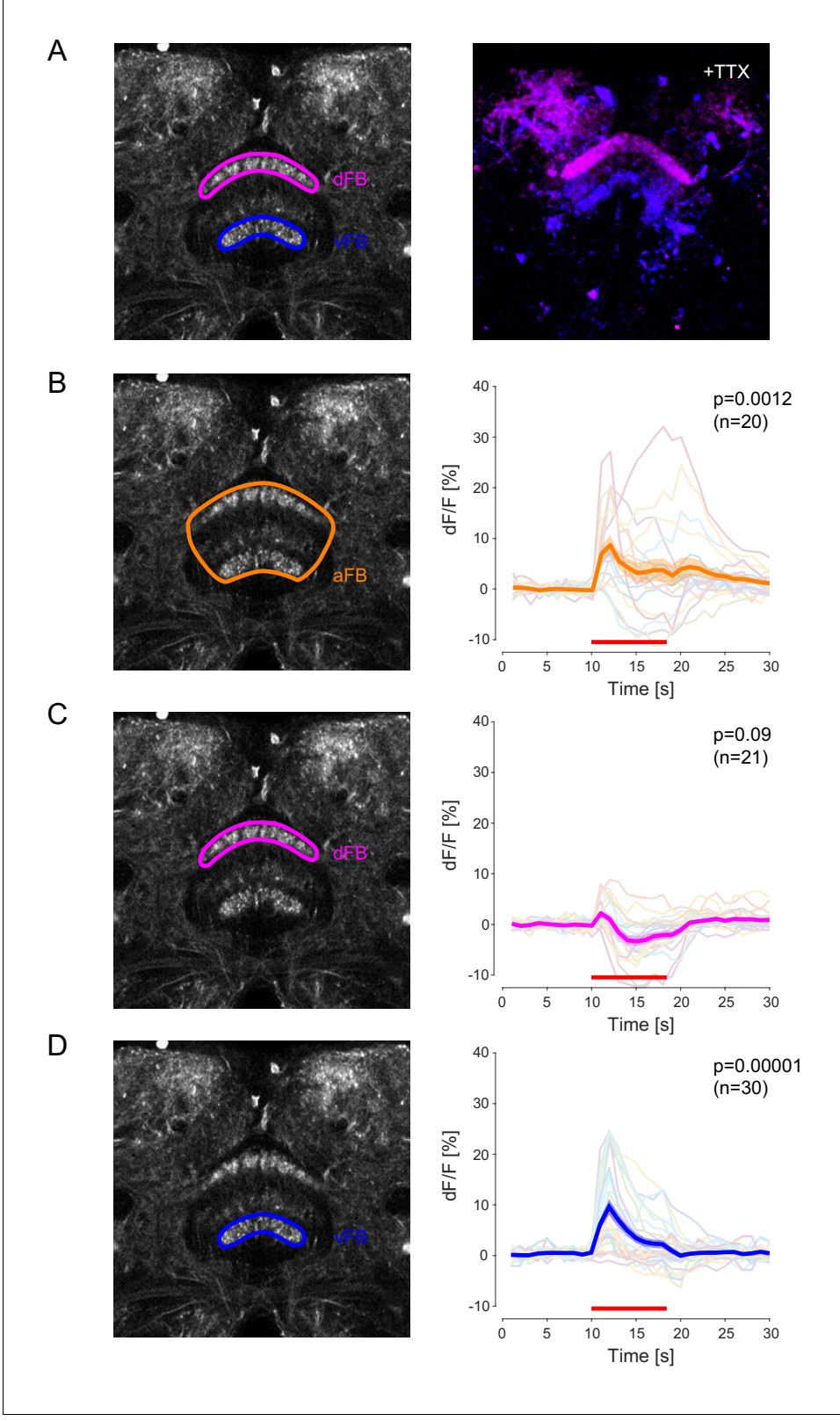

**Figure 3.** FB neurons provide an excitatory input to DAN-aSP13. (**A**) (left) Expression pattern of *104y-GAL4 > UAS-Chrimson88-tdTomato* with depicted dFB (magenta) and vFB (blue) layers for local activation with DMD. (right) Excitatory response of dFB (magenta) or vFB (blue) layers in the presence of 20 uM tetradotoxin (TTX) upon activation of dFB or vFB, respectively. The calcium response pattern evoked by stimuli was calculated by the

*Figure 3 continued on next page*

*Figure 3 continued*

correlation of determination. (**B**) (left) Expression pattern of *104y-GAL4 > UAS-Chrimson88-tdTomato* with depicted 104y FB neurons (orange) for local activation with DMD. (right) Normalized calcium levels (dF/F) in DAN-aSP13 upon local activation of 104y FB neurons (*104y-GAL4 > UAS-Chrimson88-tdTomato, R58E02-LexA > LexAop-GCamP6s*). DAN-aSP13 activity in individual flies is shown in colored thin lines and the mean trace is shown in a thick orange line with SEM indicated by shaded area. (**C**) (left) Expression pattern of *104y-GAL4 > UAS-Chrimson88-tdTomato* with depicted dFB layer (magenta) for local activation with DMD. (right) Normalized calcium levels (dF/F) in DAN-aSP13 upon local activation of dFB (*104y-GAL4 > UAS-Chrimson88-tdTomato, R58E02-LexA > LexAop-GCamP6s*). DAN-aSP13 activity in individual flies is shown in colored thin lines and the mean trace is shown in a thick magenta line with SEM indicated by shaded area. (**D**) (left) Expression pattern of *104y-GAL4 > UAS-Chrimson88-tdTomato* with depicted vFB layer (blue) for local activation with DMD. (right) Normalized calcium levels (dF/F) in DAN-aSP13 upon local activation of vFB (*104y-GAL4 > UAS-Chrimson88-tdTomato, R58E02-LexA > LexAop-GCamP6s*). DAN-aSP13 activity in individual flies is shown in colored thin lines and the mean trace is shown in a thick blue line with SEM indicated by shaded area. (**B, C, D**): Red line indicates the time of the light stimulus. P value in all panels represents the probability that the mean dF/F of pre-stimulation (10 s) and the mean dF/F during stimulation has the same median across flies (tested by Wilcoxon rank sum test, sample size indicated with n value).

DOI: https://doi.org/10.7554/eLife.42786.009

The following figure supplement is available for figure 3:

**Figure supplement 3.** Activation of the mFB has no effect on DAN-aSP13 activity.

DOI: https://doi.org/10.7554/eLife.42786.010

collections for specific GAL4 driver lines with expression in these two cell types. We identified one sparse GAL4 line with expression in the dFB neurons (*R23E10*) and one in the vFB neurons (*VT036875*). Since the expression pattern of the VT036875-GAL4 line also includes a class of DAN-β'1 neuron, we combined this line with PAM-GAL80 to restrict its expression to vFB neurons only (*VT036875-GAL4, R58E02-GAL80*). In addition, we generated a split line (*SS057264-GAL4*) (*Pfeiffer et al., 2010*; *Luan et al., 2006*) with the expression restricted to the vFB neurons only (*Figure 4—figure supplement 1B, C, D, E*).

As previously reported, thermogenetic activation of dFB over a 24 hr time period resulted in an increased amount of sleep (*R23E10-GAL4 > UAS-TrpA1*) (*Figure 4A*). Sleep increase was displayed mainly during the day time since males sleep almost continuously during night. We found that activation of vFB (*VT036875-GAL4 > UAS-TrpA1; VT036875-GAL4 > UAS-TrpA1, R58E02-GAL80; SS057264-GAL4 > UAS-TrpA1*) also robustly induced day time sleep (*Figure 4B,C,D*) in comparison to the genetic control (*pBDP-GAL4 > UAS-TrpA1*) (*Figure 4—figure supplement 1G*). Loss of night time sleep observed upon prolonged vFB activation with VT036875-GAL4 (*Figure 4B*) was likely caused by co-activation of wake promoting DAN-β'1 neurons (*Sitaraman et al., 2015a*) (*Figure 4—figure supplement 1C*). Accordingly, males upon activation with the same line while excluding DAN-β'1 neurons displayed a higher level of both day and night sleep (*Figure 4C*, *Figure 4—figure supplement 1D*). Similarly, acute optogenetic activation of the dFB and vFB neurons with CsChrimson for 1 hr also significantly enhanced the amount of sleep (*R23E10-GAL4 > UAS -CsChrimson; VT036875-GAL4 > UAS-CsChrimson; VT036875-GAL4 > UAS-CsChrimson, R58E02-GAL80; SS057264-GAL4 > UAS-CsChrimson; 104y-GAL4 > UAS-CsChrimson*) in comparison to the genetic control (*pBDP-GAL4 > UAS-CsChrimson*) but to a lesser degree when DAN-β'1 neurons were co-activated (*Figure 4—figure supplement 1F*).

To test whether these neurons provide an excitatory input on DAN-aSP13s, we activated either dFB (*R23E10-GAL4 > UAS-Chrimson88*) or vFB (*VT036875-GAL4 > UAS-Chrimson88; VT036875-GAL4 > UAS-Chrimson88, R58E02-GAL80; SS057264-GAL4 > UAS-Chrimson88*) neurons and monitored calcium levels in DAN-aSP13s. Since specific DANs innervate MB lobes in well-defined discrete areas, we used a broad PAM-DAN GAL4 driver (*R58E02-LexA > LexAop-GCamP6s*). To monitor activity specifically in DAN-aSP13s upon activation of FB neurons we focused on the region at the tip of the MBγ lobe (*Figure 3—figure supplement 1A*). Activation of dFB neurons did not elicit calcium changes in DAN-aSP13s (*Figure 4A*) however, activation of vFB neurons elicited a robust increase of calcium levels in DAN-aSP13s (*Figure 4B,C,D*). These data show that neurons in the vFB provide excitatory input onto DAN-aSP13s.

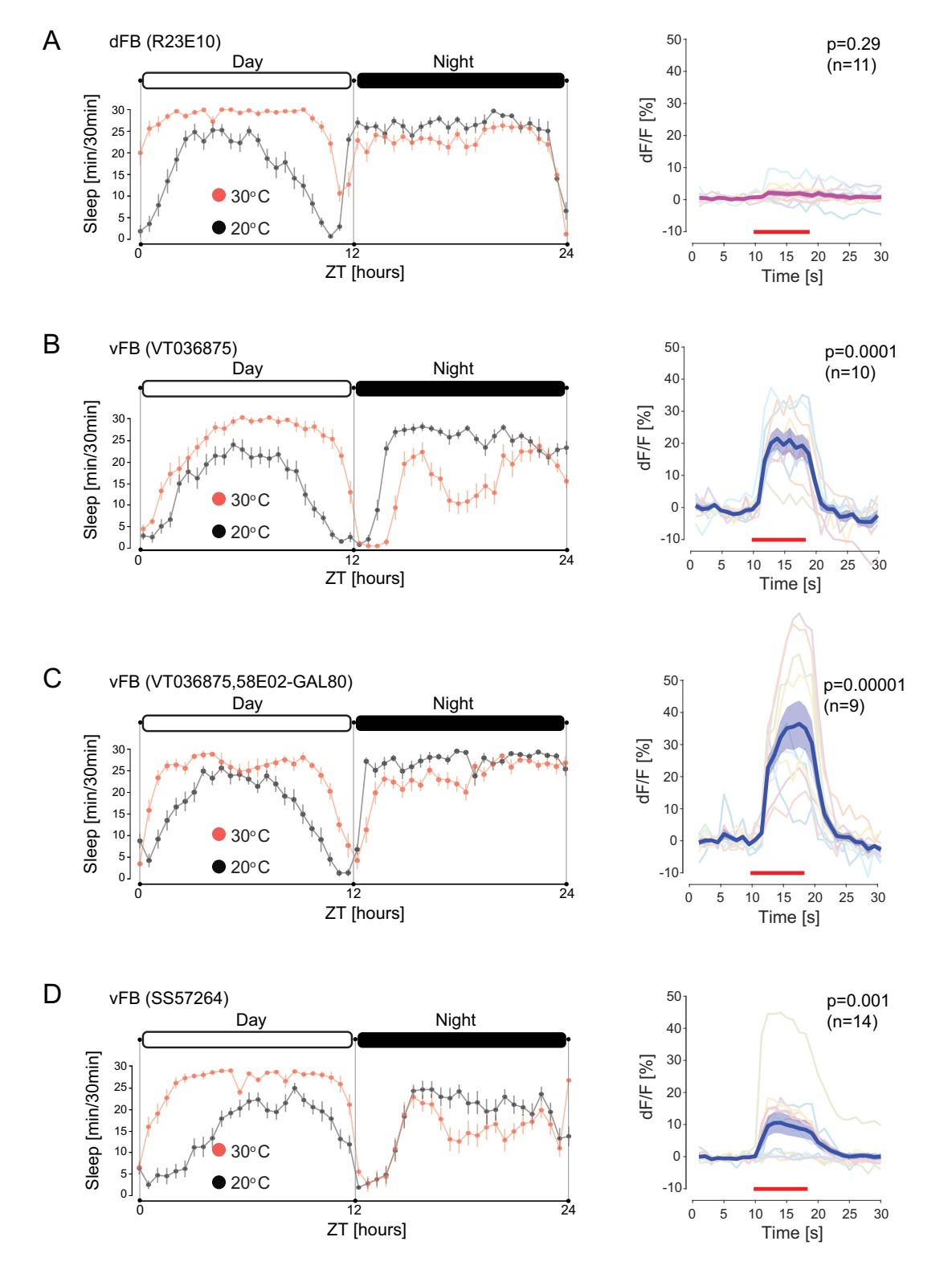

**Figure 4.** Sleep promoting vFB neurons activate DAN-aSP13. (**A**) (left) Sleep profile of males (*R23E10-GAL4 > UAS-TrpA1*) upon activation of dFB neurons (red, n = 16) and control males with dFB neurons not activated (black, n = 16). Sleep time was plotted in 30 min bins. White and black areas indicate 12 hr light and dark periods, respectively. (right) Normalized calcium levels (dF/F) in DAN-aSP13 upon activation of dFB (*R23E10-GAL4 > UAS-Chrimson88, R58E02-LexA > LexAop-GCamP6s*). DAN-aSP13 activity in individual flies is shown in colored thin lines, and the mean trace is shown in
*Figure 4 continued on next page*

*Figure 4 continued*

thick magenta line with SEM indicated by shaded area. (**B**) (left) Sleep profile of males (*VT03687-GAL4 > UAS-TrpA1*) upon activation of vFB neurons (red, n = 16) and control males with vFB neurons not activated (black, n = 16). Sleep time was plotted in 30 min bins. White and black areas indicate 12 hr light and dark periods, respectively. (right) Normalized calcium levels (dF/F) in DAN-aSP13 upon activation of vFB (*VT036875-GAL4 > UAS-Chrimson88, R58E02-LexA > LexAop-GCaMP6s*). DAN-aSP13 activity in individual flies is shown in colored thin lines, and the mean trace is shown in thick blue line with SEM indicated by shaded area. (**C**) (left) Sleep profile of males (*VT036875-GAL4 > UAS-TrpA1, R58E02-GAL80*) upon activation of vFB neurons (red, n = 16) and control males with vFB neurons not activated (black, n = 16). Sleep time was plotted in 30 min bins. White and black areas indicate 12 hr light and dark periods, respectively. (right) Normalized calcium levels (dF/F) in DAN-aSP13 upon activation of vFB (*VT036875-GAL4 > UAS-Chrimson88, R58E02-GAL80, R58E02-LexA > LexAop-GCaMP6s*). DAN-aSP13 activity in individual flies is shown in colored thin lines, and the mean trace is shown in thick blue line with SEM indicated by shaded area. (**D**) (left) Sleep profile of (*SS57264-GAL4 > UAS-TrpA1*) males upon activation of vFB neurons (red, n = 16) and control males with vFB neurons not activated (black, n = 16). Sleep time was plotted in 30 min bins. White and black areas indicate 12 hr light and dark periods, respectively. (right) Normalized calcium levels (dF/F) in DAN-aSP13 upon activation of vFB (*SS57264-GAL4 > UAS-Chrimson88, R58E02-LexA > LexAop-GCaMP6s*). DAN-aSP13 activity in individual flies is shown in colored thin lines, and the mean trace is shown in thick blue line with SEM indicated by shaded area. (**A–D**) (right panels) Red line indicates the time window of the light stimulus. P value represents the probability that the mean dF/F of pre-stimulation (10 s) and the mean dF/F during stimulation has the same median across flies (tested by Wilcoxon rank sum test, sample size indicated with n value).

DOI: https://doi.org/10.7554/eLife.42786.011

The following figure supplement is available for figure 4:

**Figure supplement 1.** Specific class of vFB neurons acutely induces sleep.

DOI: https://doi.org/10.7554/eLife.42786.012

## vFB neurons mediate LTM consolidation

Consistent with the functional connectivity results, optogenetic activation of dFB neurons (*R23E10-GAL4 > UAS*-CsChrimson) between 5–7 hr after a 1 hr training period did not lead to LTM consolidation. However, activation of either a combination of dFB and vFB neurons (*104y-GAL4 > UAS*-CsChrimson) or just the vFB neurons (*VT036875-GAL4 > UAS-CsChrimson; VT036875-GAL4 > UAS-CsChrimson, R58E02-GAL80; SS057264-GAL4 > UAS*-CsChrimson) in that same time window fully consolidated STM to LTM (*Figure 5A*, *Figure 5—figure supplement 1A*). In addition, silencing of dFB neurons (*R23E10-GAL4 > UAS-Shi^ts*) in the period between 7–10 hr after the onset of a 6 hr training period did not affect LTM persistence, while silencing of a combination of dFB and vFB or just the vFB neurons (*VT036875-GAL4 > UAS-Shi^ts; SS057264-GAL4 > UAS-Shi^ts and 104y-GAL4 > UAS-Shi^ts*) in the same time window strongly impaired LTM consolidation (*Figure 5B*, *Figure 5—figure supplement 1B*). These results show that a class of sleep-promoting neurons in the vFB layer of the 104y-FB expression pattern is necessary and sufficient to consolidate LTM during post-learning sleep. dFB neurons, although they promote sleep, appear to have no role in courtship LTM consolidation.

## Discussion

The activity of DAN-aSP13s, which is essential for courtship memory acquisition (*Keleman et al., 2012*), is also necessary during a discrete post-learning time window for LTM consolidation (*Krüttner et al., 2015*). Because neuronal reactivation occurs during sleep in rodents (*Wilson and McNaughton, 1994*; *Smith, 2001*), we hypothesized that post-learning activation of DAN-aSP13s involves a sleep-dependent mechanism. Using behavioral analysis and neuronal activity monitoring and perturbation approaches, we have shown here that DAN-aSP13s display an increased activity in freely behaving animals during sleep after a prolonged learning experience. We demonstrated that this sleep is necessary for LTM consolidation, and it can be mediated by a specific class of sleep promoting neurons in the ventral layer of the FB (vFB). These vFB neurons consolidate courtship LTM in a discrete time window and provide an excitatory input to DAN-aSP13s. Thus, the data we present here provide a causal link between sleep promoting neurons in the vFB, post-learning activation of dopaminergic neurons, and LTM consolidation.

Based on our data, we propose the following model for sleep-dependent consolidation of courtship LTM in *Drosophila* (*Figure 6*). During a prolonged learning experience, γKCs and DAN-aSP13s are repeatedly activated by olfactory and behavioral cues presented by an unreceptive female, respectively. Whereas prolonged wakefulness leads to an increase in homeostatic sleep drive in

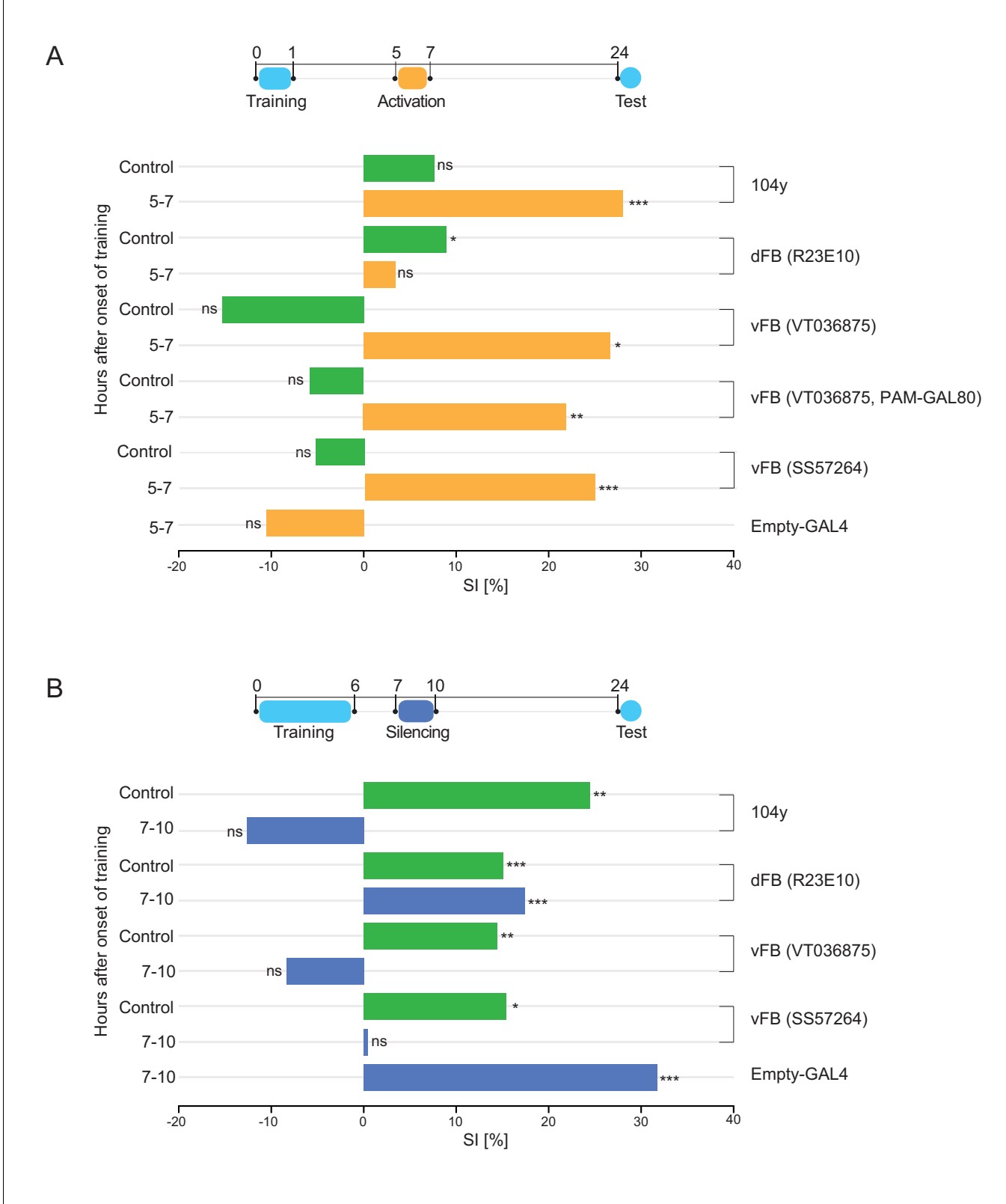

**Figure 5.** Sleep promoting vFB neurons are sufficient and necessary for LTM consolidation. (A) SIs of males of indicated genotypes tested 24 hr after 1 hr training with a mated female and activation at the specific time interval with CsChrimson (orange bars). SI of control males with relevant neurons not activated (green bar). P value is for Ho SI = 0; ***p<0.001, **p<0.01, *p<0.05, ns p>0.05. Permutation test. (B) SIs of males of indicated genotypes

*Figure 5 continued on next page*

*Figure 5 continued*

tested 24 hr after training for 6 hr with a mated female and silencing with shi[ts] at the specific time interval (dark blue bars). SI of wild type control males with DAN-aSP13 active (green bar). P value is for Ho SI = 0; ***p<0.001, **p<0.01, *p<0.05, ns p>0.05. Permutation test.

DOI: https://doi.org/10.7554/eLife.42786.013

The following figure supplement is available for figure 5:

**Figure supplement 5.** Courtship Indices (CIs) of males that had undergone treatment according to *Figure 5A and B*.

DOI: https://doi.org/10.7554/eLife.42786.014

ellipsoid body neurons that in turn is conveyed to dFB (*Liu et al., 2016*), we hypothesize that an extended learning experience generates a learning-dependent sleep drive that is transmitted to vFB. In turn, the vFB neurons enhance sleep after learning and provide an excitatory input back on DAN-aSP13s. We believe that one potential site of a learning-dependent sleep drive are the MB neurons since they have been implicated in both memory formation (*Heisenberg et al., 1985*) and sleep regulation (*Sitaraman et al., 2015b*). Dopamine released upon DAN-aSP13 reactivation stimulates molecular processes in γKCs (*Krüttner et al., 2015*) that are different from those engaged during STM acquisition and involve protein synthesis that is essential for LTM formation and persistence.

Dopaminergic pathways are thought to convey information about whether an experience is rewarding or punishing and thus, worth remembering (*Gais et al., 2002*; *Schultz, 2010*). Post-learning neuronal activity of the dopaminergic hippocampal inputs from the Ventral Tagmental Area (VTA) has been implicated in the consolidation of fear memory in rodents. Interestingly, this activity is critical during a discrete time window after learning (*Rossato et al., 2009*). Post-learning activity of the VTA dopamine neurons has been also implicated in the reactivation during sleep of the hippocampal cells involved earlier in encoding of the spatial experience (*Gomperts et al., 2015*). Here we show that post-learning activation of DAN-aSP13s mediates the consolidation of courtship LTM in

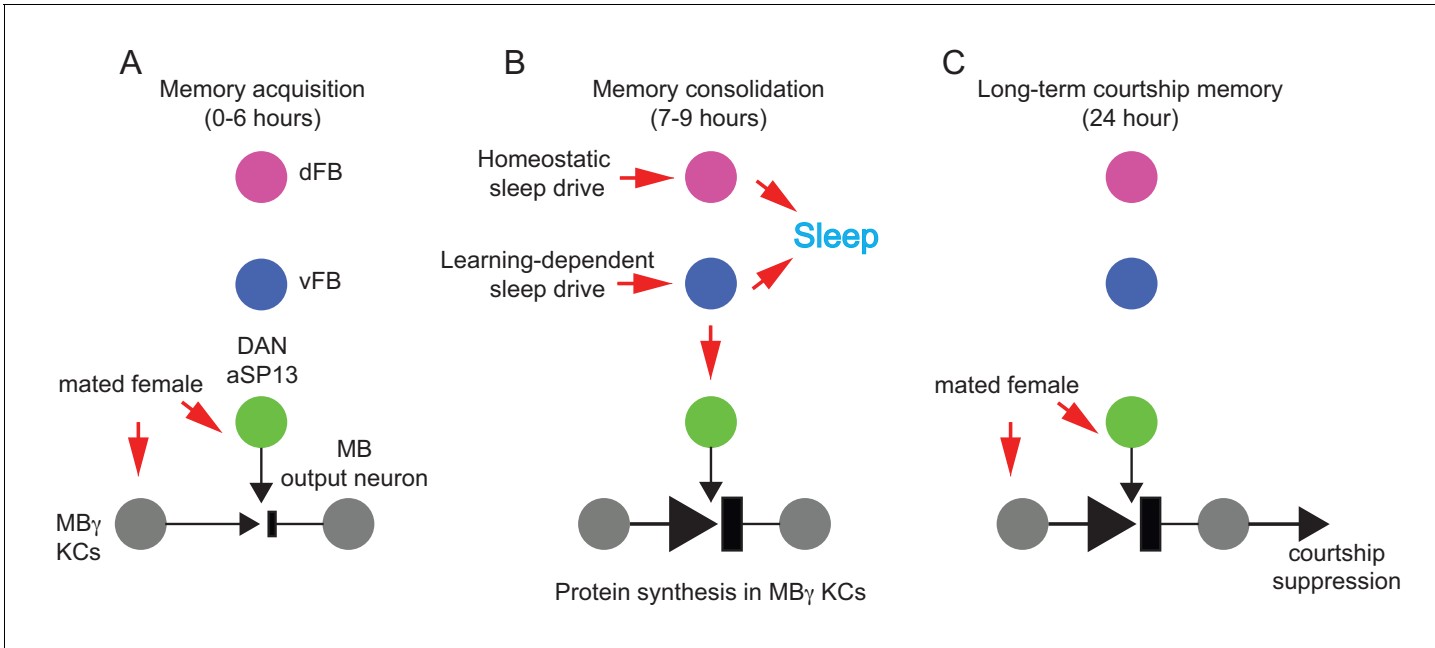

**Figure 6.** Post-learning activation of DAN-aSP13 neurons mediates LTM consolidation. (**A**) The MBγ and DAN-aSP13s are repetitively activated during 6 hr training by the olfactory and behavioral cues presented by a mated female, respectively. (**B**) Males display an enhanced amount of sleep after training for LTM. Enhanced sleep is mediated by the vFB neurons in response to a learning induced sleep drive while the remaining amount of sleep is regulated by dFB neurons in response to homeostatic sleep drive. Only vFB neurons activate DAN-aSP13s. Dopamine released as a result of DAN-aSP13s activation stimulates molecular processes in the γKCs neurons that involve synthesis of new proteins essential for LTM memory persistence. (**C**) Subsequently, experienced males suppress their courtship towards mated females for 24 hr or longer.

DOI: https://doi.org/10.7554/eLife.42786.015

*Drosophila*. We propose that reactivation during sleep of the dopamine neurons that were previously active during memory acquisition ensures that spurious experiences are not admitted into LTM storage and thus only experiences that are either sufficiently salient or persistent become long-lasting memories. Specifically, reactivation of DAN-aSP13s during post-learning sleep enhances reactivation of the γKCs and cognate MBON-M6, which together with DAN-aSP13s form a recurrent circuit necessary for courtship memory acquisition (*Zhao et al., 2018*). We consider two hypotheses to account for this selectivity. During sleep, vFB neurons might selectively reactivate only the relevant DANs, or alternatively, they might activate all DANs but only the relevant subset is able to consolidate LTM. The selective-reactivation model would require some marker to distinguish which DANs were activated, whereas the selective-consolidation model would require a marker in the synapses of the γKCs that were earlier active during memory acquisition, for example translational regulator Orb2, which regulates translation upon neuronal activity during LTM consolidation (*Krüttner et al., 2015*).

Activity of dopaminergic neurons regulates sleep-wake states in animals, including flies (*Dzirasa et al., 2006*). Artificial activation of DAN-aSP13s has been shown to increase wakefulness (*Sitaraman et al., 2015a*). In contrast, the data we present here imply that activation of vFB neurons, although they activate DAN-aSP13s, do not promote wakefulness. These results suggest that activation of DAN-aSP13s by vFB neurons during sleep is qualitatively different from a direct optogenetic or thermogenetic activation used in previous studies. One potential explanation is that post-learning sleep involves activation of the vFB circuit, which provides both excitatory stimulus to DAN-aSP13s and inhibitory input to motor neurons. Another possibility is that post-learning sleep activates a subset od DAN-aSP13s that do not affect wakefulness.

We have identified here a class of sleep-promoting neurons in the ventral layer of FB that are distinct from the well-studied sleep-promoting neurons in the dorsal layer of FB, which regulate sleep homeostasis (*Donlea et al., 2011*; *Donlea et al., 2014*; *Pimentel et al., 2016*). Given that vFB neurons enhance sleep and activate DAN-aSP13s for LTM consolidation, whereas dFB neurons are neither necessary nor sufficient for LTM consolidation, we hypothesize that dFB and vFB neurons promote distinct components of sleep that have different functions. Homeostatic sleep is thought to facilitate memory encoding by downscaling synaptic weights and clearing metabolites from the brain accumulated during wakefulness (*Bushey et al., 2011*; *Xie et al., 2013*; *Tononi and Cirelli, 2014*). In contrast, the function of learning-dependent sleep might be to facilitate memory consolidation by strengthening synaptic connections that were engaged earlier during memory acquisition (*Frank, 2012*). Thus, the co-operation of homeostatic- and experience-dependent sleep would facilitate optimal conditions for learning new information and, if appropriate, incorporating it into long-term storage.

Recent studies have implied that sleep in flies, as in humans and rodents, exhibits sleep stages characterized by distinct electrophysiological signatures (*Yap et al., 2017*; *van Alphen et al., 2013*). Interestingly, the signature of sleep that is induced by activation of the dFB neurons seems to have a simpler oscillatory pattern, recorded by local field potentials, than sleep that is induced by activation of the FB neurons comprising both dFB and vFB neurons (*Yap et al., 2017*). Thus, these data support our hypothesis that dFB and vFB neurons promote sleep with different properties and likely different functions.

It is thought that sleep evolved in animals that are capable of complex learning which requires selective attention (*Kirszenblat and van Swinderen, 2015*). Courtship learning is a multisensory form of learning that requires selective attention of a male to associate multiple learning cues presented by the mated female with the outcome of his own behavior (*Kirszenblat and van Swinderen, 2015*). Accordingly, studies in bees have shown that sleep affects a complex form of learning such as spatial memory but has no role in the simple learning paradigm of proboscis extension (*Vorster and Born, 2015*). Hence, it would be interesting to investigate whether post-learning sleep is involved in the consolidation of other types of memory in *Drosophila*, such as the well-studied Pavlovian olfactory associative learning whereby animals associate an individual learning cue with a behavioral contingency.

In this work, we establish a functional link between a novel class of sleep-promoting neurons in the FB, post-learning reactivation of dopaminergic neurons and consolidation of courtship LTM. Moreover, our data suggest that sleep promoting vFB neurons mediate a learning-dependent regulation of sleep that is distinct from the homeostatic control which is facilitated by dFB neurons

(*Liu et al., 2016*). Thus, we uncover a causal link between sleep-mediated neuronal reactivation and LTM consolidation in *Drosophila*. In addition, we establish courtship LTM in *Drosophila* as a tractable model to investigate the mechanisms that link learning-dependent sleep, neuronal reactivation and LTM consolidation.

# Materials and methods

## Key resources table

| Reagent type (species) or resource | Designation | Source or reference | Identifiers | Additional information |
|---|---|---|---|---|
| Genetic reagent (*D. melanogaster*) | UAS-FLP.PEST; lola >> luc | DOI: 10.1073/pnas.1706608114 | | Obtained from Michael Rosbash lab |
| Genetic reagent (*D. melanogaster*) | MB315B-GAL4 | DOI: 10.7554/eLife.04577 | | Obtained from Gerry Rubin lab |
| Genetic reagent (*D. melanogaster* | UAS-myr::smGFP | DOI: 10.1534/genetics.110.119917 | BDSC ID: #32197 | Obtained from Gerry Rubin lab |
| Genetic reagent (*D. melanogaster*) | Dopr1(attP) | DOI: 10.1038/nature11345 | | Obtained from Barry Dickson lab |
| Genetic reagent (*D. melanogaster*) | VT005526-LexA | DOI: 10.1016/j.celrep.2015.05.037 | | |
| Genetic reagent (*D. melanogaster*) | 104y-GAL4 | DOI: 10.1126/science.1202249 | FlyBase ID: FBti0072312 | Bloomington Stock Center |
| Genetic reagent (*D. melanogaster*) | UAS-CsChrimson-mVenus | DOI: 10.1038/nmeth.2836 | BDSC ID: #55134, #55136 | Obtained from Vivek Jayaraman lab |
| Genetic reagent (*D. melanogaster*) | LexAop2-Cs Chrimson-mVenus | DOI: 10.1038/nmeth.2836 | BDSC ID: #55139 | Obtained from Vivek Jayaraman lab |
| Genetic reagent (*D. melanogaster*) | pBDP-GAL4 | DOI: 10.7554/eLife.04577 | | Obtained from Gerry Rubin lab |
| Genetic reagent (*D. melanogaster*) | UAS-TrpA1 | DOI: 10.1101/gad.1278205 | | |
| Genetic reagent (*D. melanogaster*) | LexAop-Shi^{ts} | PMID: 12745632 | | |
| Genetic reagent (*D. melanogaster*) | LexAop-myr::smGFP | DOI: 10.1534/genetics.110.119917 | BDSC ID: #32203 | Obtained from Gerry Rubin lab |
| Genetic reagent (*D. melanogaster*) | UAS-GCaMP6s | DOI: 10.1073/pnas.1703090115 | | Obtained from David Anderson lab |
| Genetic reagent (*D. melanogaster*) | R58E02-LexA | DOI: 10.1073/pnas.0803697105 | BDSC ID: #52740 | Obtained from Gerry Rubin lab |
| Genetic reagent (*D. melanogaster* | UAS-Chrimson88-tdTomato | DOI: 10.1073/pnas.1703090115 | | Obtained from David Anderson lab |
| Genetic reagent (*D. melanogaster*) | LexAop2-Syn21-opGCaMP6s | DOI: 10.1073/pnas.1703090115 | | Obtained from David Anderson lab |
| Genetic reagent (*D. melanogaster*) | R23E10-GAL4 | DOI: 10.1073/pnas.0803697105 | BDSC ID: 49032 | Obtained from Gerry Rubin lab |
| Genetic reagent (*D. melanogaster*) | VT036875-GAL4 | DOI: 10.1101/198648 | VDRC ID: 203402 | |
| Genetic reagent (*D. melanogaster*) | R58E02-Gal80 | DOI: 10.1073/pnas.1703090115 | Flybase ID FBtp0097258 | Obtained from Glenn Turner lab |
| Genetic reagent (*D. melanogaster*) | UAS-Shi^{ts} | PMID: 12745632 | | |

*Continued on next page*

*Continued*

| Reagent type (species) or resource | Designation | Source or reference | Identifiers | Additional information |
|---|---|---|---|---|
| Genetic reagent (*D. melanogaster*) | *SS57264-GAL4* | This paper | | generated by combining the *VT045282-GAL4p65adz(attP40)* and *VT020829-ZpGAL4dbd(attP2)* lines. |
| Chemical Compund, drug | Luciferin | GOLDBIO | Cat. LUCK | |
| Chemical Compound, drug | Tetradotoxin | abcam | Cat. ab120054 | |
| Software, algorithm | Permutation Test | PMID: 10355520 | | |
| Software, algorithm | GraphPad Prism | GraphPad Prism (https://graphpad.com) | Version 7 | RRID:SCR_015807 |
| Software, algorithm | FIJI | FIJI (https://imagej.net/Fij) | | RRID:SCR_002285 |

## *Drosophila* culture conditions

Flies for behavioral experiments were reared in vials with standard cornmeal food at 25°C, or as indicated, at 60% humidity in a 12 hr light:12 hr dark cycle. Detailed information regarding specific strains and genotypes is provided in the Key resources table section.

## Behavior

Courtship conditioning was performed as described (*Siwicki and Ladewski, 2003*). Briefly, solitary males aged for 5–6 days after eclosion were placed for training (during day time as indicated) in food chambers for 1 hr (STM) or 6 hr (LTM) either with (trained) or without (naïve) a single mated female. After training each male was recovered, allowed to rest for 30 min or 24 hr and tested with a fresh mated female. Tests were performed in 10 mm diameter chambers and videotaped for 10 min (Prosilica GT cameras, Allied Vison Technologies). Automated video analysis was used to derive a courtship index (CI) for each male, defined as the percentage of time over a 10 min test period during which the male courts the female. Memory was calculated as a suppression index (SI) that is a relative reduction in the mean courtship indices of trained (CI$^+$) versus naïve (CI$^-$) populations: SI = $100*[1-CI^{train+}/CI^{train-}]$.

To monitor sleep over a 24 hr time period, male flies were individually inserted into 65 mm glass tubes containing standard fly food, loaded into TriKinetics *Drosophila* activity monitors (DAM), and housed under 12 hr light:12 hr dark schedule (*Donelson et al., 2012*). Periods of inactivity lasting at least 5 min were classified as sleep. Total 24 hr sleep quantity (day time and night time sleep) was extracted from DAM system as described (*Donelson et al., 2012*).

To monitor sleep upon acute induction with csChrimson, individual males were reared at 25°C on retinal supplemented (0.1 mM) cornmeal medium in darkness for 4–5 days after eclosion. For sleep induction single males were placed in 10 mm diameter behavioral chambers in a temperature and illumination-controlled box and videotaped (Prosilica GT cameras, Allied Vison Technologies). The amount of sleep was scored manually with periods of inactivity lasting at least 5 min being considered as sleep.

For sleep deprivation, flies were subjected to intermittent mechanical perturbation method while housed in TriKinetics DAM monitors. Flies received mechanical perturbations on a horizontal shaker with a total cycle of 15 s/min delivered in eight pulses of 1–3 s each occurring intermittently at random times.

## Courtship assays

A MATLAB script (*Kamyshev et al., 1999*) (permutation test) implemented in *Keleman et al. (2012)* was used for statistical comparison of SIs between two groups. Briefly, the entire set of courtship indices for both naïve and trained flies were pooled and then randomly assorted into simulated naïve and trained groups of the same size as the original data. A SI was calculated for each of 100,000

randomly permutated data sets, and P values were estimated for the null hypothesis that learning equals 0 ($H_0$: SI = 0) or for the null hypothesis that experimental and control males learn equally well ($H_0$: SI = $SI_c$).

## Luminescence assay

Luminescence assay for detecting neuronal activity in freely behaving adult flies was modified from *Guo et al. (2017)*. Flies were starved on filter soaked in water overnight prior to training. They were transferred to fresh chambers with filter paper containing 40 mM Luciferin (GOLDBIO) in a 2% sucrose solution. For luminescence measurements after training, flies were placed into 96-well plates (Greiner Bio-one) with 40 mM Luciferin in a 2% sucrose solution. CLARIOstar microplate reader (BMG Labtech) was used for luminescence detection. Luminescence was measured every 15 min over 16 hr. Relative Luminescence (R. Luminescence) was calculated by dividing the luminescence of the experienced and naïve groups by the luminescence of the genetic controls (males with luciferase reporter only, fed with luciferin) at every measurement time point.

## Optogenetic activation in intact animals

Single housed male flies were reared at 25°C on retinal supplemented (0.1 mM) cornmeal medium in darkness for 4–5 days after eclosion. For sleep induction or memory consolidation specific classes of neurons were optogenetically activated in behavior chambers housed in a temperature and illumination-controlled box. During activation, the behavior chamber was illuminated with 617 nm LEDs (Red-Orange LUXEON Rebel LED—122 lm; Luxeon Star LEDs, Brantford, Ontario, Canada) with a 3 mm thick diffuser between the LED and flies. The LED was driven by a customized linear current controller. 20 HZ red light was used to activate neurons with intensity varying from 20 uW/mm2 to 70 uW/mm2. The LED board was cooled by a customized liquid cooling system to maintain the temperature in the chamber.

## Optogenetic activation in explant brains

Single housed male flies were reared at 25°C on retinal supplemented (0.1 mM) cornmeal medium in darkness for 4–5 days after eclosion. Brains of the immobilized flies on ice were dissected out in saline (103 mM NaCl, 3 mM KCl, 2 mM $CaCl_2$, 4 mM $MgCl_2$, 26 mM $NaHCO_3$, 1 mM $NaH_2PO_4$, 8 mM trehalose, 10 mM glucose, 5 mM TES, bubbled with 95% $O_2$/5% $CO_2$) and mounted anterior up on the cover slip in a Sylgard-lined dish in a 20°C saline bath.

Brains were imaged with a resonant scanning two-photon microscope with near-infrared excitation (920 nm, Spectra-Physics, INSIGHT DS DUAL) and a 25x objective (Nikon MRD77225 25XW). The microscope was controlled by ScanImage 2015.v3 (Vidrio Technologies). Images of brain volume were acquired with ~ 157 μm x 157 μm field of view at 512 × 512 pixels resolution. Each frame and each volume was sampled by 42 frames with 3 μm per step, at approximately 1 Hz volume rate. Times series of volume images were acquired and the excitation power for calcium imaging ~ 12 mW.

For the optogenetic activation, the light-gated ion channel Chrimson88 was activated with a 660 nm LED (M660L3 Thorlabs) coupled to a digital micromirror device (DMD) (Texas Instruments DLPC300 Light Crafter) and combined with the imaging light path using a FF757-DiO1 dichroic (Semrock). On the emission side, the primary dichroic was Di02-R635 (Semrock), the detection arm dichroic was 565DCXR (Chroma), and the emission filters were FF03-525/50 and FF01-625/90 (Semrock). Photostimulation light was delivered in a pulse train that consisted of six 8 s or 15 s pulses (100% duty cycle during each pulse) with a 30 s inter-pulse interval. The light intensity was ~ 0.6 mW/mm2, as measured using Thorlabs S170C power sensor.

All image data were analyzed off-line. Region of interest (ROI) of DAN-aSP13 at mushroom body γ5 compartment was manually defined in Fiji or CircuitCatcher (a customized python program by Daniel Bushey) and the average GCaMP signal intensity within the DAN-aSP13 ROI was taken as the calcium activity of DAN-aSP13. Time series calcium activity of DAN-aSP13 (f(t)) was extracted from the image data and then analyzed with customized Matlab (MathWorks) programs. The normalized Calcium activity of DAN-aSP13 dF/F is defined as:

$$\mathrm{dF/F} = \frac{f(t) - F0}{F0}$$

where the f(t) is the calcium signal intensity and the F0 is the mean F of the first 10 s of the image session before optogenetic activation. dF/F traces of six stimulation were aligned to the LED onset and averaged to represent the DAN-aSP13 activity upon neuron activation. To determine the connectivity from the activated neuron to DAN-aSP13, the mean dF/F during 10 s pre-stimulation and the mean dF/F during stimulation were taken as baseline activity and stimulated activity in a fly. Groups of baseline activities and stimulated activities of different flies were tested with student's test or Wilcoxon rank sum test to determine if optogenetic activation had evoked significant calcium activity changes in DAN-aSP13 against the hypothesis that the baseline activity and stimulated activity were the same level. p value smaller than 0.05 was taken as the criteria of connectivity (either inhibitory or excitatory).

## Correlation of determination

Correlation of determination ($r^2$) was used to measure the correlation between the time series calcium trace (f(t)) of each voxel and a predicted model. Here we used the voltage driving the stimulation LED (V(t)) as the model.

The calcium trace of each voxel was firstly normalized to z-score, calculated with the following function

$$Z(t) = \frac{f(t) - \mu}{\sigma}$$

Where $\mu$ is the mean and $\sigma$ is the standard deviation estimated of the f(t).

Then by a linear fit of the model V(t), we have the predicted calcium response Zp (t). Correlation coefficient (r) between Z(t) and Zp(t) was calculated by:

$$r = \frac{cov(Z(t), Zp(t))}{\sigma Z * \sigma Zp}$$

where $cov(Z(t), Zp(t))$ is the covariance of Z(t) and Zp(t), σZ and σZp are the standard deviation of Z(t) and Zp(t).

The correlation of determination ($R^2$) was simply the square of the r. $R^2$ is by definition in the range of 0 to 1. The higher the value is, the higher is the correlation (both positive and negative) between a voxel's calcium trace and stimuli. The $r^2$ indexes of a whole brain volume are projected to a 16-bit image stack, which demonstrates the calcium response pattern evoked by stimuli.

Multicolor Flip-out (MCFO) was performed according to the protocol described in *Nern et al. (2015)*.

Immunostaining was performed according to a protocol described in *Zhao et al. (2018)*.

## Acknowledgements

We thank Kristin Henderson for help with the behavioral assays, Kelley Lee and Oz Malkesman for help with the multicolor flip-out, Wyatt Korff and Nan Chen for help with sleep assays, Gerry Rubin for sharing unpublished FB lines, and Habib Bukhari for help with the figures. We thank Ulrike Heberlein, Barry Dickson and Gerry Rubin for critical comments on the manuscript. This work was supported by Howard Hughes Medical Institute-Janelia Research Campus.

## Additional information

### Funding

| Funder | Author |
|---|---|
| Howard Hughes Medical Institute | Krystyna Keleman |

The funders had no role in study design, data collection and interpretation, or the decision to submit the work for publication.

## Author contributions
Ugur Dag, Investigation—behavioral experiments, Data analysis, Writing—original draft, Writing—review and editing; Zhengchang Lei, Investigation—Most of the optogenetic experiments, Data analysis, Writing—original draft, Writing—review and editing; Jasmine Q Le, Investigation—setting up DMD experiments, Data analysis, Original draft editing; Allan Wong, Daniel Bushey, Investigation-optogenetic experiments, Data analysis; Krystyna Keleman, Conceptualization, Supervision, Funding acquisition, Investigation, Writing—original draft, Project administration, Writing—review and editing

## Author ORCIDs
Ugur Dag ⓘD https://orcid.org/0000-0001-6937-5722
Zhengchang Lei ⓘD https://orcid.org/0000-0002-6475-5010
Jasmine Q Le ⓘD https://orcid.org/0000-0003-4159-8830
Allan Wong ⓘD https://orcid.org/0000-0002-8492-2162
Daniel Bushey ⓘD http://orcid.org/0000-0001-9258-6579
Krystyna Keleman ⓘD https://orcid.org/0000-0003-2044-1981

## Decision letter and Author response
Decision letter https://doi.org/10.7554/eLife.42786.021
Author response https://doi.org/10.7554/eLife.42786.022

---

# Additional files

## Supplementary files
• Supplementary file 1. Supplementary Tables. Table S1. LTM does not depend on the circadian time of training Suppression indices (SIs) of naïve (train-) and experienced (train+) males of the indicated genotypes according to *Figure 1—figure supplement 1E*, tested in single-pair assays with mated females as trainers and testers. Courtship indices (CIs) are shown as median of $n$ males and dispersion of the data as interquartile range (*IQR*). $P$ values determined by permutation test for the null hypothesis that learning equals 0 ($H_0$: SI = 0) or for the null hypothesis that experimental and control males learn equally well ($H_0$: SI = $SI_c$). Table S2. *dopR1* mutant court mated females more than wild-type males Courtship indices (CIs) of naïve males of the indicated genotypes according to *Figure 1—figure supplement 1G* during 6 hr training with mated female are shown as median of $n$ males and dispersion of the data as interquartile range (*IQR*). $P$ values determined by permutation test for the null hypothesis that CIs of both groups are equal ($H_0$: $CI_{wt}$ = $CI_{DopR1}$). Table S3. *dopR1* mutant males do not learn Suppression indices (SIs) of naïve (train-) and experienced (train+) males of the indicated genotypes according to *Figure 1—figure supplement 1H*, tested in single-pair assays with mated females as trainers and testers. Courtship indices (CIs) are shown as median of $n$ males and dispersion of the data as interquartile range (*IQR*). $P$ values determined by permutation test for the null hypothesis that learning equals 0 ($H_0$: SI = 0) or for the null hypothesis that experimental and control males learn equally well ($H_0$: SI = $SI_c$). Table S4. Day-time sleep deprivation between 7–9 hr impairs LTM Suppression indices (SIs) of naïve (train-) and experienced (train+) males of the indicated genotypes, sleep deprived as denoted in *Figure 2A*, tested in single-pair assays with mated females as trainers and testers. Courtship indices (CIs) are shown as median of $n$ males and dispersion of the data as interquartile range (*IQR*). $P$ values determined by permutation test for the null hypothesis that learning equals 0 ($H_0$: SI = 0) or for the null hypothesis that experimental and control males learn equally well ($H_0$: SI = $SI_c$). Table S5. Night-time sleep deprivation does not impair LTM Suppression indices (SIs) of naïve (train-) and experienced (train+) males of the indicated genotypes, sleep deprived as denoted in *Figure 2*-figure supplement A, tested in single-pair assays with mated females as trainers and testers. Courtship indices (CIs) are shown as median of $n$ males and dispersion of the data as interquartile range (*IQR*). $P$ values determined by permutation test for the null hypothesis that learning equals 0 ($H_0$: SI = 0) or for the null hypothesis that experimental and control males learn equally well ($H_0$: SI = $SI_c$). Table S6. Silencing of DAN-aSP13 between 7–9 hr impairs LTM Suppression indices (SIs) of naïve (train-) and experienced (train+) males of the indicated genotypes with DAN-aSP13 silenced as denoted in *Figure 2B*, tested in single-pair assays with mated

females as trainers and testers. Courtship indices (CIs) are shown as median of *n* males and dispersion of the data as interquartile range (*IQR*). *P* values determined by permutation test for the null hypothesis that learning equals 0 ($H_0$: SI = 0) or for the null hypothesis that experimental and control males learn equally well ($H_0$: SI = $SI_c$). Table S7. Sleep induction between 5–7 hr consolidates STM to LTM Suppression indices (SIs) of naïve (train-) and experienced (train+) males of the indicated genotypes and sleep induced as denoted in *Figure 2C*, tested in single-pair assays with mated females as trainers and testers. Courtship indices (CIs) are shown as median of *n* males and dispersion of the data as interquartile range (*IQR*). *P* values determined by permutation test for the null hypothesis that learning equals 0 ($H_0$: SI = 0) or for the null hypothesis that experimental and control males learn equally well ($H_0$: SI = $SI_c$). Table S8. DAN-aSP13 activation between 5–7 hr consolidates LTM Suppression indices (SIs) of naïve (train-) and experienced (train+) males of the indicated genotypes with DAN-aSP13 activated as denoted in *Figure 2D*, tested in single-pair assays with mated females as trainers and testers. Courtship indices (CIs) are shown as median of *n* males and dispersion of the data as interquartile range (*IQR*). *P* values determined by permutation test for the null hypothesis that learning equals 0 ($H_0$: SI = 0) or for the null hypothesis that experimental and control males learn equally well ($H_0$: SI = $SI_c$). Table S9. Activation of vFB neurons between 5–7 hr consolidates LTM Suppression indices (SIs) of naïve (train-) and experienced (train+) males of the indicated genotypes with vFB activated as denoted in *Figure 5A*, tested in single-pair assays with mated females as trainers and testers. Courtship indices (CIs) are shown as median of *n* males and dispersion of the data as interquartile range (*IQR*). *P* values determined by permutation test for the null hypothesis that learning equals 0 ($H_0$: SI = 0) or for the null hypothesis that experimental and control males learn equally well ($H_0$: SI = $SI_c$). Table S10. Silencing of vFB neurons between 7–10 hr impairs LTM Suppression indices (SIs) of naïve (train-) and experienced (train+) males of the indicated genotypes with vFB silenced as denoted in *Figure 5B*, tested in single-pair assays with mated females as trainers and testers. Courtship indices (CIs) are shown as median of *n* males and dispersion of the data as interquartile range (*IQR*). *P* values determined by permutation test for the null hypothesis that learning equals 0 ($H_0$: SI = 0) or for the null hypothesis that experimental and control males learn equally well ($H_0$: SI = $SI_c$).
DOI: https://doi.org/10.7554/eLife.42786.016

• Supplementary file 2. Fly genotypes. Specific fly genotypes used in all main and supplementary figures.
DOI: https://doi.org/10.7554/eLife.42786.017

• Source code 1. Source code for analysis of calcium traces.
DOI: https://doi.org/10.7554/eLife.42786.018

• Transparent reporting form
DOI: https://doi.org/10.7554/eLife.42786.019

### Data availability

Source data files have been provided for Figure 1—figure supplement 1 and 2, Figure 2, Figure 2—figure supplement 1 and 2 and Figure 5 and Figure 5—figure supplement 1.

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
