## [Decision Letter]

Thank you for submitting your article "Neuronal reactivation during post-learning sleep consolidates long-term memory in *Drosophila*" for consideration by *eLife*. Your article has been reviewed by three peer reviewers, one of whom is a member of our Board of Reviewing Editors, and the evaluation has been overseen by Ronald Calabrese as the Senior Editor. The reviewers have opted to remain anonymous.

The reviewers have discussed the reviews with one another and the Reviewing Editor has drafted this decision to help you prepare a revised submission.

Summary:

This work uses *Drosophila* to test and advance the idea that long-term memory (LTM) consolidation involves reactivation during sleep of neurons that become activated during the period when the memory is created. The authors previously showed that activation of a particular group of dopaminergic neurons in the mushroom bodies (DAN-aSP13) is necessary for courtship short term memory (STM). Using a luminescent reporter, the authors found that the activity of DAN-aSP13s were increased in male flies exposed to mated females, relative to naïve males. They also showed that if they deprived males of sleep during a critical period following training with mated females their courtship was not suppressed. The same effect occurred if they silenced DAN-aSP13s during the same time windows following training. Moreover, activating sleep during the critical period following a very short training that is ordinarily not sufficient for LTM induced suppression, and this requires activation of DAN-aSP13s. The authors also show that activation of vFB neurons in turn activates DAN-aSP13s, and this contributes to LTM consolidation.

Thus, through an elegant set of experiments, the authors demonstrate that DAN-aSP13s neurons, which are active during courtship behaviour, must also be active during sleep for LTM formation. Learning-induced sleep pressure is mediated by vFB neurons, which provide an excitatory input to DAN-aSP13s neurons. The conclusions are consistent with unexpected previous findings from the same lab that there are two temporally distinct phases of dopaminergic activity required for LTM consolidation in *Drosophila* and also with work showing multiple sleep phases in flies and mammals. The authors raise the very interesting and testable possibility that two phases of sleep in flies have different functions, while identifying a function for vFB driven sleep. Overall, this study is well done and will be of significant interest to the sleep field.

Essential revisions:

1) Together, the authors suggest that "reactivation" of DAN-aSP13s neurons during sleep is comparable to neural replay in rodents. Neural replay in rodents involves the sequential activation of neurons, that represent a behavioural sequence. Replay detection is based on the very small probability that the sequence could reactivate by chance. The same cannot be said about "reactivation" of general neural activity (no sequence). This happens during sleep all the time (e.g. up and down states, spindles) and while this is not treated as reactivation, disrupting this type of activity would likely interfere with memory consolidation. For example, is DAN-aSP13 activation or silencing specific for the training task or all learning that occurred during that period. The idea of reactivation is that it is a specific memory that is replayed. In its current state, the data of dopamine neuronal "reactivation" in the fly doesn't seem comparable to rodent replay.

In all likelihood DANs represent emotional quality associated with experience, not the experience itself. For instance experience of odor A associated with a shock or odor B associated with a shock are not distinguished at the level of DANs. Given that the work monitors DANs, and does not monitor replay of experience, it is important to clearly acknowledge that the work does not directly address a function for experience replay during sleep.

2) Please explain and provide details on how MB315B-GAL4>UAS-FLP, Lola>stop>LUC provides an activity dependent reporter. It should not be necessary to dig into the literature to understand this. Is MB315B only expressed when there is neuronal activity (how well has this been shown?) and is <ola only expressed in DAN-aSP13s?

3) Figure 1A-E. Do explain why there is a one hour delay between exposure of the mated males to females and the beginning of the luminescence measurements. Why is there a delay in the increase in activity? Also, why is the activity increased in the naïve males, albeit with a delay. The AU are a bit unclear. Is there really a >30x increase in activity in the males exposed to mated females? Why are there periods during the assays that are neither day nor night? It is not clear what would comprise such a condition.

4) It seems that the sham (naïve) males are males that are left alone rather than are exposed to virgin females. Therefore, another necessary control is to expose males to virgin females.

5) The statement that thermogenetic activation of vFB neurons with TRPA1 increases sleep is true during the day. However, as shown in Figure 4, nighttime sleep is reduced when there is increased sleep in the day. Why only these two mutants? It doesn't seem linked to exclusive vFB or dFB activation. Any comments on this?

6) The experiments monitoring DAN activity while stimulating FB neurons use a wide-spectrum DAN driver, not a specific for DAN-aSP13 as used in the earlier experiments. A sentence of two with arguments to rule out the possibility that alternative DANs being activated by vFB neurons somehow result in the same outcome is required (Despite Figure 3—figure supplement 1A).

7) The phrase "necessary and sufficient" is likely used inappropriately in at least two different parts of the text e.g. DAN neuron activity may be necessary but is certainly not sufficient for LTM consolidation – it is only sufficient when various other criteria including all the sensorimotor experiences associated with training have occurred and post training sleep has been induced. (Though I acknowledge that similar usage is common in the field see: https://www.nature.com/articles/d41586-018-05418-0.)

8) Another control experiment that could be useful is to examine potential courtship suppression after DAN neurons have been activated during sleep in naive animals that have *not* been trained. This would rule out the possibility of non-specific fly-equivalent of "anxiety," for instance being a confounding phenomenon.

---

## [Author Response]

Essential revisions:1) Together, the authors suggest that "reactivation" of DAN-aSP13s neurons during sleep is comparable to neural replay in rodents. Neural replay in rodents involves the sequential activation of neurons, that represent a behavioural sequence. Replay detection is based on the very small probability that the sequence could reactivate by chance. The same cannot be said about "reactivation" of general neural activity (no sequence). This happens during sleep all the time (e.g. up and down states, spindles) and while this is not treated as reactivation, disrupting this type of activity would likely interfere with memory consolidation. For example, is DAN-aSP13 activation or silencing specific for the training task or all learning that occurred during that period. The idea of reactivation is that it is a specific memory that is replayed. In its current state, the data of dopamine neuronal "reactivation" in the fly doesn't seem comparable to rodent replay.In all likelihood DANs represent emotional quality associated with experience, not the experience itself. For instance experience of odor A associated with a shock or odor B associated with a shock are not distinguished at the level of DANs. Given that the work monitors DANs, and does not monitor replay of experience, it is important to clearly acknowledge that the work does not directly address a function for experience replay during sleep.

We fully agree that reactivation of DAN-aSP13s as we present here, is not comparable to the sequential reactivation of neuronal activity observed during consolidation of spatial memory in rodents. We modified the text accordingly (Abstract; Introduction, second paragraph; Discussion, third paragraph).

2) Please explain and provide details on how MB315B-GAL4>UAS-FLP, Lola>stop>LUC provides an activity dependent reporter. It should not be necessary to dig into the literature to understand this. Is MB315B only expressed when there is neuronal activity (how well has this been shown?) and is <ola only expressed in DAN-aSP13s?

We provide details on how MB315B-GAL4>UAS-FLP, Lola>stop>LUC functions as an activity dependent reporter in DAN-aSP13s (subsection “DAN-aSP13 neurons are activated during post-learning sleep”, first paragraph).

3) Figure 1A-E. Do explain why there is a one hour delay between exposure of the mated males to females and the beginning of the luminescence measurements.

The 1-hour delay between training and start of the luminescence measurements reflects the time period when we separate individual males after training from trainer females and place them into luminescence reading plates. Since the time of these manipulations varies slightly between the animals, we used a common time point for all males to start the measurements (subsection “DAN-aSP13 neurons are activated during post-learning sleep”, first paragraph).

Why is there a delay in the increase in activity?

Our data suggest that activity of DAN-aSP13s is linked to the time period when sleep is enhanced after training. The gradual increase in DAN-aSP13 activity corresponds very well to the gradual sleep increase in this time window (Figure 1A and B).

Also, why is the activity increased in the naïve males, albeit with a delay.

The increase in the activity in naïve animals is of smaller amplitude and of slightly different dynamics than in trained males. We think that this activity increase is not related to learning because when we train males with virgin females which does not induce LTM, the activity increase in the experienced and naïve animals is small and identical (Figure 1—figure supplement 1B).

The AU are a bit unclear. Is there really a >30x increase in activity in the males exposed to mated females?

We performed now additional luminescence experiments. The n of animals in both groups increased roughly 2-fold. To present our data more clearly, we now calculated the Relative Luminescence (R. Luminescence) by dividing the luminescence of trained and naïve males by the luminescence of the corresponding genetic controls at every measurement time point. Our data shows that experienced males display about 0.5x increase in activity in comparison to naïve males. We provide description of R. Luminescence in the Materials and methods section (subsection “Luminescence Assay”).

Why are there periods during the assays that are neither day nor night? It is not clear what would comprise such a condition.

Our assays do not have time periods that are neither day or night. We have corrected our confusing schematics in all figures accordingly.

4) It seems that the sham (naïve) males are males that are left alone rather than are exposed to virgin females. Therefore, another necessary control is to expose males to virgin females.

We have now trained males with virgin females, which does lead to LTM when tested 24 after training with mated females, and monitored luminescence using Lola reporter in DAN-aSP13s. Males that had undergone training with virgin females displayed a small increase in luminescence which was identical to that in naïve males. These results suggest that luminescence increase in males that were trained with mated females reflects learning that leads to LTM (Figure 1—figure supplement 1B, subsection “DAN-aSP13 neurons are activated during post-learning sleep”, first paragraph).

5) The statement that thermogenetic activation of vFB neurons with TRPA1 increases sleep is true during the day. However, as shown in Figure 4, nighttime sleep is reduced when there is increased sleep in the day. Why only these two mutants? It doesn't seem linked to exclusive vFB or dFB activation. Any comments on this?

Reduction of night time sleep after prolonged sleep induction is observed mainly in Figure 4B. This particular GAL4 driver in addition to vFB neurons drives expression in DAN-β’1s (Figure 4—figure supplement 1C) which were implicated in promoting wakefulness (Sitaraman et al., 2015). Accordingly, males upon activation of vFB with the same line while silencing DAN-β’1 (Figure 4—figure supplement 1D) displayed a higher level of both day and night sleep (Figure 4C).

Similarly, acute optogenetic activation with CsChrimson for 1 hour significantly enhanced sleep but to a lesser degree than when DAN-β’1s were co-activated (Figure 4—figure supplement 1F) (subsection “vFB neurons promote sleep and activate DAN-aSP13s”, second paragraph). A small decrease of night sleep after sleep induction with the split line SS057264-GAL4 (Figure 4D, Figure 4—figure supplement 1E)might be the result of an increased homeostatic sleep drive to reduce sleep after prolonged sleep induction and the inability of the weaker split line to override it (split GAL4s are considered generally weaker than the standard GAL4 lines). Notably, acute sleep induction with the same line was very effective to induce sleep over short period of time (Figure 4—figure supplement 1F).

The effect of sleep increase upon thermogenetic activation seems to be specific to ventral and dorsal FB neurons. We have tested multiple FB neuronal classes in the course of this study but were able to induce sleep by activation of these two FB neuronal cell types only.

6) The experiments monitoring DAN activity while stimulating FB neurons use a wide-spectrum DAN driver, not a specific for DAN-aSP13 as used in the earlier experiments. A sentence of two with arguments to rule out the possibility that alternative DANs being activated by vFB neurons somehow result in the same outcome is required (Despite Figure 3—figure supplement 1A).

Since specific DANs innervate MB lobes in well-defined discrete areas, we used a broad PAM-DAN GAL4 driver (*R58E02-LexA>LexAop-GCamP6s*). To monitor activity specifically in DAN-aSP13 upon activation of FB neurons we focused on the region of interest (ROI) at the tip of the MBg lobe which we demarcated in Figure 3—figure supplement 1A (subsection “vFB neurons promote sleep and activate DAN-aSP13s”, last paragraph).

7) The phrase "necessary and sufficient" is likely used inappropriately in at least two different parts of the text e.g. DAN neuron activity may be necessary but is certainly not sufficient for LTM consolidation – it is only sufficient when various other criteria including all the sensorimotor experiences associated with training have occurred and post training sleep has been induced. (Though I acknowledge that similar usage is common in the field see: https://www.nature.com/articles/d41586-018-05418-0.)

We have introduced appropriate changes in the text (Abstract; Introduction, last paragraph; subsection “Enhancement of sleep consolidates LTM”; Discussion, first paragraph).

8) Another control experiment that could be useful is to examine potential courtship suppression after DAN neurons have been activated during sleep in naive animals that have not been trained. This would rule out the possibility of non-specific fly-equivalent of "anxiety," for instance being a confounding phenomenon.

We now activated DAN-aSP13s in naïve males during the time period when males display usually a significant amount of day sleep and when STM can be consolidated to LTM after short training. We tested their courtship with mated females after 24 hours. CIs of naïve males with DAN-aSP13 either activated or not were indistinguishable from each other but significantly different from CI of trained males when tested with mated females (Figure 2—figure supplement 1D). These results suggest that consolidation of courtship LTM is a specific effect of DAN-aSP13 activation after training (subsection “Enhancement of sleep consolidates LTM”, last paragraph).